# The Role of Mononuclear Phagocytes in the Testes and Epididymis

**DOI:** 10.3390/ijms24010053

**Published:** 2022-12-20

**Authors:** Xu Shi, Hu Zhao, Yafei Kang, Xinyi Dong, Caiqian Yu, Qinying Xie, Yonggang Duan, Aihua Liao, Donghui Huang

**Affiliations:** 1Institute of Reproduction Health Research, Tongji Medical College, Huazhong University of Science and Technology, Wuhan 430030, China; 2Department of Humans Anatomy, Tongji Medical College, Huazhong University of Science and Technology, Wuhan 430030, China; 3Shenzhen Key Laboratory of Fertility Regulation, Centre of Assisted Reproduction and Embryology, The University of Hong Kong-Shenzhen Hospital, Shenzhen 518053, China; 4Shenzhen Huazhong University of Science and Technology Research Institute, Shenzhen 518109, China

**Keywords:** mononuclear phagocyte system (MPS), macrophages, dendritic cells, testes, epididymis, immune, inflammation, male infertility

## Abstract

The mononuclear phagocytic system (MPS) is the primary innate immune cell group in male reproductive tissues, maintaining the balance of pro-inflammatory and immune tolerance. This article aims to outline the role of mononuclear macrophages in the immune balance of the testes and epididymis, and to understand the inner immune regulation mechanism. A review of pertinent publications was performed using the PubMed and Google Scholar databases on all articles published prior to January 2021. Search terms were based on the following keywords: ‘MPS’, ‘mononuclear phagocytes’, ‘testes’, ‘epididymis’, ‘macrophage’, ‘Mφ’, ‘dendritic cell’, ‘DC’, ‘TLR’, ‘immune’, ‘inflammation’, and ‘polarization’. Additionally, reference lists of primary and review articles were reviewed for other publications of relevance. This review concluded that MPS exhibits a precise balance in the male reproductive system. In the testes, MPS cells are mainly suppressed subtypes (M2 and cDC2) under physiological conditions, which maintain the local immune tolerance. Under pathological conditions, MPS cells will transform into M1 and cDC1, producing various cytokines, and will activate T cell specific immunity as defense to foreign pathogens or self-antigens. In the epididymis, MPS cells vary in the different segments, which express immune tolerance in the caput and pro-inflammatory condition in the cauda. Collectively, MPS is the control point for maintaining the immune tolerance of the testes and epididymis as well as for eliminating pathogens.

## 1. Introduction

Male reproductive tissues include the testes and epididymis, which are places for sperm generation and storage. It is well known that they exist in a unique immune environment, where the latent autoimmune response must be suppressed in order to establish an immune tolerance to sperm that appears after puberty [1]. Therefore, this tolerance is critical for the prevention of autoimmune responses against antigenic post-pubertal germ cells and anti-sperm antibody (AsAb) production [2]. Meanwhile, these tissues also need to resist pathogen invasion and infection. It was reported that infection and inflammation constitute 13–15% of all cases of male factor infertility, including epididymitis, combined epididymo-orchitis and isolated orchitis [3]. Therefore, it is crucial to maintain the balance of pro-inflammatory and immune tolerance for male fertility.

In 1972, van Furth et al. first proposed the mononuclear phagocyte system (MPS) in *humans*, which includes three types of immune cells: macrophages (Mφ), dendritic cells (DC) and their precursor cell—the monocyte [4]. Studies have shown that the DC plays a key role in maintaining immune tolerance and immune activation [5], while macrophage polarization could affect the fate of inflammatory or injured organs in *humans* and *mice* [6]. Similar to other parts, the MPS in the testes and epididymis participates in immune survillance and inflammation and regulates the complex interaction between immune toleance and activation in *humans* and *mice* [7]. Furthermore, the MPS in testes and epididymis is the main immune cell group during inflammation in *humans* and *mice* [8,9]. Therefore, the MPS occupies a very important position in testicular and epididymal immunity. In order to better understand the immune environment in the testes and epididymis and to provide a new perspective for diagnosis and treatment of male fertility, this article reviews the generation, location, classification as well as the anti-inflammatory and pro-inflammatory effects of MPS cells.

## 2. Methods

This narrative review is based on various types of basic research. In a qualitative rather than quantitative way, it describes the changes in the immune status of the male reproductive system under physiological and inflammatory conditions and corrects some previous misunderstandings, such as the mixed use of MPS cell surface markers, ultimately aiming to evaluate and interpret the latest existing research results. The authors conducted an extensive search in the PubMed and Google Scholar databases, with keywords including ‘MPS’, ‘mononuclear phagocytes’, ‘testes’, ‘epididymis’, ‘macrophage’, ‘Mφ’, ‘dendritic cell’, ‘DC’, ‘halo cell’, ‘TLR’, ‘immune’, ‘inflammation’, ‘polarization’, and ‘male reproductive system’. Halo cells have been postulated to be either lymphocytes or monocytes and were thus included in the search terms [10,11]. In addition, each author independently reviewed the articles and references in order to find all content that met the requirements. The selected articles were verified by three independent authors and discussed within the team.

The main focus of this article is the inflammatory immunity of the testes and epididymis. Other tissues of the male reproductive tract such as prostate or other pathological conditions such as tumors are not covered.

## 3. The Occurrence of the MPS in Testes and Epididymis

Monocytes in the *human* body stem from hematopoietic stem cells (HSCs) through intermediate progenitor cells [12]. Myeloid progenitors (CMPs) that differentiate from HSCs first differentiate into monocyte–dendritic cell progenitor cells (MDPs) and granulocyte–monocyte progenitor cells (GMPs) in *humans* and *mice* [13,14]. MDP generates two subsets, partly into common monocyte progenitors (cMoPs), which then produce classical monocytes, which in turn transform into macrophages. The remaining MDPs transform into DCs through common DC progenitors (CDPs)—pre-DCs. The hierarchical relationship of DCs eventually produces conventional DC1 (cDC1), conventional DC2 (cDC2) and plasmacytoid DC (pDC). In addition, CDPs can differentiate into pDC through alternative differentiation pathways [13]. GMPs then produce a subset of “neutrophil-like” monocytes, which share a typical monocyte marker (Ly6Chi) with MDP-derived monocytes in *mice* [14]. The differentiation direction of monocytes from this source is unclear, thereby causing disagreement in the literature about how monocytes occur.

*Human* monocytes are divided into three types: classical (CD14^++^ CD16^−^), intermediate (CD14^++^ CD16^+^) and non-classical (CD14^+^ CD16^++^) monocytes [15]. Classical monocytes can differentiate into intermediate and non-classical monocytes (9); 90% of *human* blood monocytes are classical monocytes, while only 10% are non-classical monocytes [16].

There seem to be two subsets similar to *human* monocytes in the blood of *mice*, but the markers are not exactly the same [15]. In *mice*, “classical” monocytes (previously called inflammatory monocytes, or GR-1+ monocytes) are characterized by the combination of surface markers Ly6Chi CX3CR1int CCR2^+^ CD62L^+^ CD43low, while “non-classical” monocytes (previously called patrolling monocytes, or GR-1^−^ monocytes) are defined as Ly6Clow CX3CR1hi CCR2low CD62L^−^ [17]. The intermediate monocyte is defined as Ly6C^++^ CD43^++^ [15].

For *humans* and *mice*, non-classical monocytes and classical monocytes respectively tend to produce non-classical macrophages (M2 macrophage) under physiological conditions and classical macrophages (M1 macrophage) under pathological conditions [15]. *Human* M1 macrophages express high levels of MHC II, CD68 and co-stimulatory molecules CD80 and CD86, while various subtypes of M2 macrophages express different markers [18]. In *mouse* testes, under physiological conditions, CD43hi CCR2^−^ CX3CR1^+^ monocytes (equivalent to *human* non-classical monocytes) act as a replenishment bank for the non-classical M2 macrophage [12,19,20]. In the pathological situation of *mouse* orchitis caused by viral or bacterial infection or autoantibodies, monocytes (Ly6C^+^ CCR2^+^ CX3CR1low, equivalent to *human* classical monocytes) expressing CCL2 receptors (i.e., CCR2^+^) in the blood flow into the testes in response to inflammatory stimuli and then differentiate into classical M1 macrophages [19,21] (Figure 1).

It was believed that during the process of embryonic hematopoiesis, some tissue-resident macrophage populations are sown and are used for independent self-maintenance in adulthood in *humans* and *mice* [22,23,24,25]. In *mouse* testes, it was previously believed that interstitial macrophages (MCSFR^+^ CD64hi MHC II^−^) originated from the embryonic yolk sac and existed at birth, while peritubular macrophages (MCSFRlow CD64low MHC II^+^) only originated from bone marrow progenitor cells, appearing only two weeks after birth [26]. However, the current understanding does not support this case. Wang et al. attest that *mouse* interstitial macrophages and peritubular macrophages come from yolk sac erythro-myeloid progenitors, embryonic hematopoiesis, and nascent neonatal monocytes in newborns [27]. Among them, embryonic CD206^−^ MHC II^−^ macrophages are derived from yolk sac macrophages, while interstitial CD206^+^ MHC II^−^ macrophages and peritubular CD206^−^ MHC II^+^ macrophages are derived from *mice* fetal liver-derived monocytes [28]. In adults, macrophages maintain their own numbers independently under physiological conditions and do not rely on bone marrow hematopoietic precursors [27]. The most likely situation is that CD206^+^ MHC II^−^ macrophages differentiate into CD206^−^ MHC II^+^ macrophages to fill the niche of macrophages [28]. Once infection occurs, the bone marrow-derived circulating monocytes are recruited to the testes and epididymis and differentiate into M1 macrophages [27]. Serre, V. et al. found that monocytes in *rat* epididymis could increase in a segment-specific manner with age [29]. Presently, the source of testicular or epididymal macrophages in embryos and adulthood is not fully understood, and the situation may be more complicated in the *human* body.

## 4. The Characteristic of the MPS in the Testes and Epididymis

### 4.1. DCs in the Testes and Epididymis

DCs are a heterogeneous cell population. As mentioned earlier, *human* DCs can be divided into one pDC and two cDC groups according to their source. cDCs have a typical dendritic appearance while pDCs lack dendrites and are reminiscent of plasma-like cells [30]. *Mouse* cDC is characterized by Lin^−^ MHC Ⅱ^+^ CD11c^+^ and can be divided into two subgroups, cDC1 and cDC2, as mentioned above [31]. The expression profiles and functions of *human* CD141^+^ and CD1C^+^ genes are similar to *mouse* cDC1s (CD8α^+^ CD103^+^) and cDC2s (CD11b^+^ CD172a^+^), respectively. Therefore, Guilliams suggested that in the unified naming scheme, *human* CD141^+^ DC can be called cDC1, and *human* CD1c ^+^ DC can be called cDC2 [16,30]. Between them, *human* cDC1s constitute a minority, which can be understood as a mature phenotype, and they express TLR3 and 8, mainly playing an antigen presentation function [31,32]. Meixlsperger et al. confirmed that in humanized *mice*, after synthetic dsRNA stimulates TLR3, cDC1s not only produce IL-12, but also produce a large amount of IFN-α, and after adding an adjuvant that promotes the maturation of cDC1s, cDC1s can induce CD4 ^+^ T cell responses [33,34]. cDC2 is the main subgroup, which can be understood as an immature phenotype that is mainly responsible for inducing T helper cell type responses [31,35]. *Human* cDC2 expresses Toll-like receptors (TLR) 1–8 and secrete IL-12, TNFα, IL-8 and IL-10 after being stimulated [31,32]. The maturation and activation state of DC is considered to be the control point for inducing peripheral tolerance and autoimmunity.

The activation state of DCs is crucial for determining the outcome of the anti-inflammatory and pro-inflammatory balance. Immature DCs have limited ability to activate T cells [36,37]. They are present in peripheral tissues as a sentinel of the immune system and respond to inflammatory stimuli (such as TNF-α) or microbial products (such as lipopolysaccharide (LPS)). These stimulus signals induce DCs to mature and migrate to secondary lymphoid organs [37]. On the other hand, mature DCs are T cell stimulants [36].

#### 4.1.1. DCs in the Testes

##### The Classification and Distribution of DCs in the Testes

Testes and epididymis research is mainly focused on *mice* and *rats*. DCs in the testes of normal *mice* show an immature phenotype with a high-level of endocytosis capacity and a downregulation of MHC molecules, co-stimulatory molecules CD80, CD86 and CD40 and chemokine receptors (e.g., CCR7), thus endowing them with antigen-presenting ability [38,39]. Research by Rival et al. suggested that *rat* testicular DCs and DCs from testicular draining lymph nodes (TLN) do not activate lymphocytes under physiological conditions, which indicates that testicular DCs are already in a state of T cell tolerance [40]. It is generally considered that under the stimulation of inflammation or pathogens, DCs turn into mature DCs. The expression levels of co-stimulatory receptors and MHC II on the cell surface increase, and this type of mature DC migrates to the draining lymph nodes, thereby triggering antigen-specific T cell responses [41,42].

DCs were located only in the interstitial tissue of *rat* testes under normal conditions [43]. De Rose et al. used Lin^−^, HLA-DR^+^ and either CD11c^+^ or CD123^+^ to label macaque testicular DCs, with a median of 0.22% [44]. More recently, research by Winnall et al. showed that in the *primate* testes, DCs account for 0.7% of the total number of testicular white blood cells [45]. Because both DCs and macrophages are located in the basal region of the epithelium and have obvious morphological similarities, they share same immune markers to a large extent. Rival C et al. demonstrated that the specific labeling of *rat* DCs with Ox62 and CD11c can help distinguish DCs and macrophages in the testes, and the total number of DCs can be measured by stereoscopic analysis [43].

##### The Function of DCs in the Testes

I. Immature DCs in the testes mainly play a role in limiting the autoimmune response

Immature DCs mainly exert their regulatory function through immunosuppression. Previous studies have found that in *humans*, the lack of expression of co-stimulatory molecules such as CD80, CD86 and CD40 and MHC II, which are mainly located in the cell membrane, hinders the clonal expansion of T cells, thereby reducing the possibility of antigen-specific immune responses in the testicular seminiferous epithelium, and particularly, the non-expression of MHC II antigen may be a mechanism to protect sperm from autoimmune attack under physiological conditions [46,47,48]. However, in *rat* testes, the expression of MHC II, CD80, and CD86 may not be the main factor regulating the function of immature DC, while CCR7, CC2R and CD70 may be, which we will discuss in the next paragraph [40,49]. DC in *rat* testes is unable to stimulate the proliferation of naive T cells, indicating that it takes the tolerance state in the physiological testicular microenvironment [43,50,51,52]. Under physiological conditions, *mouse* testicular DCs also secrete pro-inflammatory cytokines such as IL-10 and IDO [53,54]. Gao et al. showed that Sertoli cells may play a role in promoting the differentiation of these immature DCs in *mouse* testes—the evidence of which is the downregulation of co-stimulatory molecules on the surface of DCs, reduced production of pro-inflammatory cytokines, and the inhibition of T-cell proliferation as well as the promotion of Treg development during co-culture with peritubular Sertoli cells [55].

It is generally believed that immature DCs are tolerogenic and mature DCs are immunogenic. New research suggests that there exists an intermediate state between mature and immature DCs. Studies by Menges et al. showed that in an experimental autoimmune encephalomyelitis *mouse* model, stimulating DCs with TNF-α can induce an incomplete maturity (semi-mature) state of DCs [56]. Repeated injections of these DCs and TNF-α into *mice* simultaneously can induce the production of Treg. Treg production can then produce IL-10, which has been considered to inhibit experimental autoimmune encephalomyelitis. These DCs already express a large number of MHC II and co-stimulatory molecules, which is more like an incomplete mature/semi-mature DC [56,57]. As mentioned above, “immature” DCs in *rat* testes and testicular draining lymph nodes under physiological conditions do not lack MHC II and co-stimulatory molecules and cannot stimulate naïve T cell proliferation in in vitro experiments [40,43,58]. On the contrary, inducing T cell anergy and producing Treg requires the presence of co-stimulatory molecules in DCs [59,60]. Based on these observations, we suggest that although DCs express MHC II and co-stimulatory molecules, DCs in normal *mouse/rat* testes do not exert an immune-promoting function. Therefore, at this time, they should be called “semi-mature” DC rather than “immature” DC, as this implies the immunity here requires finer adjustments. This sub-population provides a direction for further research. However, in the remainder of this review, we will still use the term “immature” DC since that is widely considered to be the majority population of the testes in the physiological state.

II. Mature DCs play a major role in inflammatory response in the testes

Acute isolated orchitis is rare, and more commonly, about 95% patients are co-infected with epididymitis [61]. Acute orchitis often occurs at the same time as epididymitis; therefore, it is also called acute epididymo-orchitis. The most frequent causes of orchitis mainly include bacterial/viral infection (caused by ascending urethra/blood infections, respectively) and autoimmunity, although the proportion of these three causes is still unclear [61,62]. The pathogens of epididymitis include sexually transmitted pathogens (more common in those less than 35 years of age) and *Escherichia coli* (*E. coli*) (more than 35 years of age). The most classic example of viral orchitis is mumps orchitis. Furthermore, autoimmune orchitis is defined as an autoimmune attack on the testes, characterized by the presence of specific AsAb, and its incidence may exceed current knowledge [63,64,65]. Due to the limited access to tissue samples, the research of immunopathological mechanisms has been largely performed in animal models instead of the *human* body. Moreover, animal models could give a better understanding of the pathophysiology and molecular pathways despite those limitations. The animal models corresponding to these three causes are animal models of bacterial epididymo-orchitis (*E. coli*/Chlamydia trachomatis), models mimicking systemic infection and inflammation, as well as the experimental autoimmune orchitis (EAO) model. The *mouse* EAO model, which is a favorite among researchers, was pioneered by Tung et al. and was induced in *rodents* by active immunization with syngeneic testicular homogenate (TH) in incomplete or complete Freund’s adjuvant (CFA) followed by injection of inactivated Bordetella pertussis (Bp) bacteria or Bp toxin [66,67].

Duan et al. demonstrated a similar distribution pattern of CD11c+ DCs in *human* testes with chronic inflammation [68]. The increase in the number of DCs in the testes of EAO *rats* may be attributed to the recruitment of new precursor cells and the in situ differentiation of DCs [69,70]. An analysis carried out by Rival et al. regarding co-stimulatory molecules and chemokine receptors in *rat* shows that in chronic orchitis, DCs are in a state of migration and have partially matured [40]. In *mice*, when EAO occurs, immature DCs that maintain immune privilege are activated by autoantigen and are induced to mature DCs, promoting the development of Th1 cells or cytotoxic T cells, thereby producing higher levels of cytokines, leading to the occurrence of EAO and causing severe damage to the testes [71,72,73]. Research by Rival et al. found that compared with control DCs, testicular DCs isolated from EAO *rats* significantly enhanced the proliferation of naïve T cells [40]. Based on the *rat* model, Bhushan et al. speculated that testicular DCs may take up spermatic antigens from the impaired seminiferous tubules to undergo immunogenic maturation and subsequently travel to the TLN via the lymphatic system [74]. Alternatively, testicular DCs might prime naïve T cells in situ, which may be the model of how DCs induce the specific immunity model of orchitis [74].

Since the control group and EAO *rat* DCs express MHC II, the percentages and intensity levels of CD80 and CD86 are similar. A surface marker that may indicate DC maturity is CCR7, since the level of CCR7 mRNA in the testicular DCs of EAO *rats* was significantly higher than that of the control group [40]. Randolph et al. demonstrated that the upregulation of CCR7 expression is the key step for the entry of DCs into lymph nodes and its homing to T cell and B cell regions [75]. Moreover, CCR7 can also regulate other functions of DCs, such as endocytosis, cell survival, and maturation (68). In addition, Rival et al. found that in the EAO testes, the expression of CCR7 mRNA in DCs increased while the expression of CCR2 decreased [40]. Therefore, the expression level of chemokine receptors, such as CCR7 and CCR2, rather than CD80, CD86, or MHC II, can be used to characterize inflammatory and steady-state DCs in *rodent* testes. In addition, as reported by Sanchez et al., in vivo experiments have shown that the ‘novel’ co-stimulatory molecule CD70 in *rat* DC is also an upregulated molecule during inflammation [49].

In addition, mature DCs (CD103^+^ MHC II^+^ CD80^+^ CD86^+^, i.e., cDC1) isolated from the EAO *rat* testes and draining lymph nodes can activate T cells. Bioactive IL-12p70 produced by mature DC can bias activated T cells in favor of an inflammatory Th1 response, and IL-23, which shares the p40 subunit with IL-12, is a crucial cytokine for the polarization of Th17 cells [40,58,76,77]. A study by Duan et al. further proved that cytokine IL-23 and Th17 cells produced by DCs and macrophages supporting cytokines (TGFβ1, IL-6, IL-21 and IL-22) might facilitate the development of *human* chronic orchitis and may affect Th17 cells under inflammatory condition [68]. Therefore, we presume that DCs are related to the occurrence of orchitis, and mature DCs have an important pro-inflammatory effect in the induction and development of immune response.

For DCs, the state of DCs (immature/semi-mature/mature) determines whether to recognize antigens to exert immune effects, or to ignore sperm antigens to further maintain local immune tolerance (Figure 2). Then, the functional transformation of DC has fine regulation on testicular immunity. Here, there exist semi-mature DCs, that is, immature DCs that highly express co-stimulatory molecules. The current role is unclear, thus offering future directions for research on this kind of cell.

#### 4.1.2. DCs in the Epididymis

The incidence of acute epididymitis is approximately 290 per 100,000 men per year [78]. In most cases, epididymitis is caused by infection. Using modern molecular biology diagnostic methods, the pathogen can be detected in 87% of patients who have not used antibiotics. *E. coli* is the main pathogen (56%)—usually due to the retrograde progression of urethral pathogens and sexually transmitted bacterial infections [64,79]. Animal models for studying epididymitis include the intraductal *E. coli* epididymitis model, the Chlamydia trachomatis epididymitis model, and models mimicking systemic infection and inflammation (which include animal models of systemic viral disease and LPS-induced inflammation model) [66]. Since patients with epididymitis are strictly prohibited from undergoing epididymal biopsy to prevent the spread of pathogens, it is difficult to obtain specimens of inflamed epididymis. Thus, the current research on epididymis is mainly based on animal experiments.

##### The Classification and Distribution of DCs in the Epididymis

As mentioned above, *human* and mouse cDCs can be divided into two subgroups, namely cDC1 and cDC2, and this is also true in the epididymis. In *mouse* epididymis, based on the expression of CD103 and CD11b, CD64^−^ CD24^+^ CD11c^+^ MHC II^+^ CD11b^−^ CD103^+^ are cDC1 and CD64^−^ CD24^+^ CD11c^+^ MHC II^+^ CD11b^+^ CD103^−^ are cDC2, and there is an extra CD103^+^ CD11b^+^ subgroup [80]. The MPS in *rodent* epididymis expresses both macrophage and dendritic cell markers in the traditional sense; thus, it is sometimes difficult to distinguish between the two cells. CD64^+^ CD11c^−^ (macrophage) and CD64^−^ CD11c^+^ (DC) can assist in the labeling of these two cells [81,82].

The distribution of epididymal DCs (eDCs) in different segments is different. Voisin et al. found that in *mouse* epididymis, cDC1 subgroups accounted for 0.35% and 0.20% of the living cells in caput and cauda, respectively, while cDC2s account for 0.50% and 0.10% [83]. It has also been confirmed through immunofluorescence that in *mouse* epididymis, the network of peritubular DCs marked by CD11c is denser in the initial segment (IS) than in the more distal segments. This distribution may be related to the maturation of sperm mainly in the proximal end of the epididymis (caput) and storage at the most distal end (cauda) [80,83,84,85]. In addition, a DC subset (CD103^+^ CD11b^+^) that also expresses both cDC1/2 markers was also found in *mouse* epididymis [76]. Da Silva labeled *mouse* DC with CD11c and observed that the DCs at the bottom of the epididymal epithelium have multiple slender dendrites projecting into the epithelium, as well as a subset of DCs strictly present in the interstitium [80]. In addition, a subset of *murine* interstitial DCs was found. Thus, the subset does not express traditional DC marker CD11c, but expresses CD206 and does not have the stellar shape of CD11c cells, which further proves the heterogeneity of eDCs [80]. Duan et al. believed that similar to the testes, in *human* epididymitis, DCs are mainly immature DCs (CD1a^+^ DC, CD11c^+^ cDC, and CD209^+^ DC) under physiological conditions, while pDC and mature DCs are only found in chronic epididymitis [53,86]. In *human* epididymis, unlike in *mouse* epididymis where a dense network of CD11c^+^ DCs are localized in the epithelium, the number of CD11c^+^ DCs was low under physiological conditions. Likewise, the majority of CD11c^+^ DCs are identified in the interstitial compartment of the proximal epididymis but not in the epididymis epithelium [53]. When inflammation occurs, the number of CD11c^+^ IL-23p19^+^ DCs in inflamed epididymis is increased and concentrated in the epididymis epithelium. Additionally, a group of CD123^+^ pDCs arrange as clusters around the small vessels or as single cells and an increased number of CD1a^+^, CD209^+^ immature DCs and CD83^+^ mature DCs is observed [53].

##### The Function of DCs in the Epididymis

In the epididymis, it is known that CD103^+^ DCs in the *mouse* epididymis express high levels of co-stimulatory molecules, while CD103^−^ DCs do not express or underexpress co-stimulatory molecules [80,87,88]. This suggests that eDCs also participate in maintaining the balance between anti-inflammatory and pro-inflammatory effects.

I. The role of eDCs in promoting immune tolerance

eDCs can participate in the establishment and maintenance of immune tolerance to mature sperm expressing new self-antigens. Da Silva et al. observed that in the *mice* IS, most epithelial cells are in direct contact with immature or semi-mature eDCs and extend dendrites into dozens of tightly connected epitheliums. These tight junctions constitute the so-called blood–epididymis barrier (BEB) [80]. In addition, the lumen of these segments is relatively narrow, thus increasing the possibility of sperm interacting directly with epithelium, thus establishing the tolerance of the immune system to sperm antigens and male fertility [80]. Smith et al. found that after epididymal efferent duct ligation (EDL), semen from the *mouse* testes was prevented from entering the IS of the epididymis, leading to extensive apoptosis of the proximal epididymal epithelial cells. DCs, however, quickly respond to the destruction caused by EDL by phagocytosing the apoptotic epithelial cells and their fragments, thus maintaining the integrity of the epididymal epithelium tight junctions [85].

Pierucci-Alves et al. believe that TGF-β signaling in DCs is required for immunotolerance to sperm in the *mouse* epididymis [82]. Pierucci-Alves et al. In addition believe that TGF-β signaling in DCs can maintain epididymal immune tolerance by keeping DCs in a passive, immature, or non-functional state [7]. The lack of TGF-β signaling can lead to severe epididymal inflammatory lesions, which suggests that TGF-β signaling in DC is a necessary factor for epididymal non-inflammatory homeostasis [7].

In the above-mentioned study by Voisin, a special cDC1/2 marker double positive CD103 ^+^ CD11b ^+^ DC subgroup was described. This subgroup may be related to the establishment of the tolerogenic state of sperm antigens transported through the *mice* epididymis [83]. This may be similar to CD103^+^ DC in the intestine by promoting the differentiation of naïve T cells into immuno-suppressive Treg cells [89,90]. B7-H3, which was originally extracted from a cDNA library of *human* DCs in 2001, exist in *mice* eDCs, and the B7-H3 level in the corpus is significantly higher than that of the caput and the cauda [91,92,93]. B7-H3 may upregulate T cell activation, promote T cell proliferation, and produce cytokines [92,94,95]. However, subsequent studies have shown that B7-H3 also has the function of mediating the negative regulation of T cells in vitro and in vivo [96,97]. It is speculated that this distribution is due to the dual task of the corpus to establish immune tolerance to descending sperm and to simultaneously resist retrograde pathogens.

II. The role of eDCs in pro-inflammatory responses

Wang et al. classified DCs in *human* and *mice* epididymis into three types: tolerogenic DC that can recognize normal sperm antigens; immunogenic DCs that can detect and clear out abnormal sperm cells and exotic pathogens; and inflammatory DCs, which can recruit Th17/Th1 cells [98]. Da Silva et al. believe that in *mouse* epididymis, unlike IS where DCs extend dendrites to capture sperm antigen to establish male reproductive tract immune tolerance, the distal area is where sperm are preserved for a long time, as the isolated CD11c ^+^ eDCs shows strong antigen presentation and cross-presentation capabilities in vitro [80]. The DCs observed in the cauda in the physiological state may be the “sentinel”. These DCs can effectively sample the contents of the lumen continuously, including all testicular antigens, exert immunity against rising pathogens, regulate inflammation, and provide supplement to the function of testicular DCs [80,99].

When chronic epididymitis occurs in *humans*, DCs produces the majority of IL-23 [53]. IL-23 is believed to play a key role in the terminal differentiation and survival of Th17 cells and is related to the development of autoimmunity and inflammation in chronic periodontitis [100]. pDC, which is only found in chronic epididymitis with markers Lin- MHC II^+^ CD303(BDCA-2)^+^ CD304 (BDCA-4)^+^ in the *human* body, is a special antiviral cell that can rapidly secrete high levels of IFN-I as the first line of defense. In addition, pDCs in the inflamed epididymis might facilitate and amplify cytotoxic T cell functions, acting as a barrier to prevent invading pathogens and to eliminate abnormal sperm cells [101].

The functions and distribution characteristics of the epididymal MPS and other cells have led to the formation of a special environment. More specifically, the caput has the characteristics of immune tolerance, while strong antigen immunity and inflammation often occur in the cauda. The epididymis needs to protect sperm that escape from the immune-exempt environment of the testes.

### 4.2. Macrophage in Testes and Epididymis

The macrophage is a very heterogeneous cell group that has two main functional states or polarization modes: classically activated or M1 macrophage (expresses CD86, MHC II) and alternatively activated or M2 macrophage (expresses CD163, CD206) [102,103]. Winnall et al. asserted that the two subtypes of macrophages in the testes were differentiated from two subtypes of monocytes (mentioned above) [19]. Macrophages are markedly plastic cells that can switch from one phenotype to another. The phenomenon is called “macrophage polarization”. Exposure of M2 macrophages to M1 signals, or vice versa, can cause differentiated macrophages to “re-polarize” or “re-program” [104].

Wang et al. showed that it is the specific microenvironment that determines the process of macrophage transformation into different phenotypes [105,106]. The unique polarization function characteristic of macrophages is manifested under the action of different cytokines and bacterial products [104]. M1 macrophages are pro-inflammatory and are polarized by LPS alone or combined with Th1 cytokines (such as IFN-γ, GM-CSF). M1 macrophages can also produce pro-inflammatory cytokines such as IL-1β, IL-6, IL-12, IL-23 and TNF-α. M2 macrophages, on the other hand, have anti-inflammatory and immunomodulatory effects and are polarized by Th2 cytokines such as IL- 4 and IL-13. M2 macrophages can produce anti-inflammatory cytokines such as IL-10 and TGF-β [104,107]. The latest research shows that cell metabolism is an important factor in determining the phenotype of macrophages. Inhibition of oxidative phosphorylation not only inhibits the state of M2 macrophages, but also actively induces polarization of M1 macrophages. Oxidative phosphorylation and activation of PGC-1β reduce the production of pro-inflammatory cytokines in M1 macrophages [108,109].

#### 4.2.1. Macrophages in the Testes

##### The Classification and Distribution of Macrophages in the Testes

In *rats*, testicular macrophages (TMs) are divided into three types: CD68^−^ CD163^+^ (M2 macrophage) are resident TMs that can differentiate into different subtypes of macrophages determined by the testicular micro-environment; CD68^+^ CD163^−^ (M1 macrophage) are the infiltrating cells that rapidly fill the testes after various inflammatory stimuli under pathological conditions (e.g., bacterial, viral, autoimmune, torsion); it follows that there is an intermediate TM of CD68^+^ CD163^+^ [20]. Under normal conditions, the majority of the TMs are resident TMs or intermediate TMs [20].

Among the testis cells isolated from *primate* (Macaca nemestrina), M2 macrophages account for the absolute majority (median 42.7%), and M1 macrophages account for 4.5% [44]. The situation is similar in *mice* under physiological conditions, as the M2 phenotype accounts for 80% of the murine TMs, while the M1 phenotype accounts for the remaining 20% [110].

TMs are mainly distributed in the peritubular space or interstitium. De Falco et al. divided *human* TMs into two types according to their distribution: peritubular TMs (pTMs) and interstitial TMs (iTMs) [111]. In *humans*, these two types of cells both express CD68 and CD163, and thus far, no markers have been found to distinguish these two TMs in *human* testes [26]. The research in *mice* is more adequate. iTMs (M-CSFR^+^ CD64hi MHCII^−^) are localized in the interstitial space between the seminiferous tubules, closely adjacent to Leydig cells and blood vessels. pTMs (M-CSFRlo CD64lo MHCII^+^), however, are close to the seminiferous tubule wall, adjacent to the peritubular muscle-like cells, showing a slender morphology [24,26,74,112]. The numbers of these two populations in *mouse* testes are similar. In addition, TMs in the same location can express different transcription profiles. CD64low MHC hi labeled *mouse* pTM expresses high levels of antigen-presenting genes, such as H2Dmb, H2Eb1 and H2K1, while CD64hi MHC low labeled pTM expresses high levels of immuno-suppressive genes, namely IL10 and Marco [26]. In addition, a CSF1R and MHC II double-negative population was also observed in *mice* testes, which may represent undifferentiated macrophages [111]. In addition to differences in location, phenotype, and appearance, the two cell sources are also different (mentioned above).

##### The Function of Macrophages in Testes

Under physiological conditions, the immunosuppressive M2 macrophages account for the majority of *mouse* and *primate* TMs. These macrophages can induce Treg to differentiate, secrete anti-inflammatory factors, and inhibit antigen presentation, and mainly play an immunosuppressive effect. The M1 macrophage plays a traditional immune-promoting function. Researchers have found that changes in the ratio of these two types of macrophages is the key to the fine balance of inflammatory response. M2 macrophages mimic the characteristics of M1 macrophages when inflammation occurs, such as increased expression of CD68 and decreased secretion of IL-10, while M1 macrophages are polarized to M2 macrophages with an increased expression of CD163, high secretion of IL-10, and low secretion of TNF-α by the interstitial fluid (IF) surrounding the TMs in *rat* testes [106]. M1 and M2 macrophage have a mutually antagonistic effect to a certain degree and jointly maintain the immune balance of the testes.

I. M2 phenotype mainly plays an immunosuppressive role

Most of the resident macrophages in the testes are classified as M2 macrophages. In *rat* testes, M2 macrophages mainly play an immunosuppressive role, while regulating spermatogenesis, steroid production, and BTB permeability of the normal testes [113,114]. There are factors that regulate the phenotype of macrophages in *rat* testes. For example, physiologically, corticosterone, PGE 2, PGI 2, and testosterone are molecules that maintain the M2 phenotype, of which corticosterone is the most important molecule [105,106]. In addition, TMs constitutively secrete a basal level of corticosterone, which can then maintain its own phenotype through paracrine/autocrine action [105]. The latest research shows that corticosterone induces the activation of AMPK and fatty acid oxidation pathways in bone marrow-derived monocytes (BMDM) and primary TM of *mice* in vitro, leading to M2 macrophage polarization. On the other hand, in vivo corticosteroid treatment can increase the ratio of M2 macrophage in a murine model of uropathogenic *E. coli* (UPEC)-elicited orchitis, and this process is partly AMPK-dependent [115]. These cytokines, hormones, and corticosteroids are widely present in the IF. Together, they affect the immunosuppressive function of TMs, thus playing an important role in the establishment and maintenance of testicular immune privilege.

The M2 phenotype, as the main type of TM under physiological conditions, mainly plays a role in maintaining homeostasis. The expression of surface molecules directly affects the anti-inflammatory effect of the M2 macrophage. A study by Wang et al. found that about half of the M2 macrophages in the *rat* testes does not express MHC II and does not express CD80 and CD86 at all. This lack of expression may cause the TM-mediated antigen presentation to T cells to be blocked, thus hindering the activation of adaptive immunity [105]. Previous studies have shown that in *human* intestines and lungs, not expressing CD80 and CD86 is considered to be one of the ways to maintain the normal homeostasis of the organs [105,116,117]. Research by Indumathy et al. showed that the frequency of M2 macrophages labeled with CD206 in the testes of adult *mice* with elevated activin A (Inha+/-) was significantly lower than that of CD206^+^ MHC II^++^ cells. Meanwhile, the frequency of CD206^+^ MHC II^−^ cells is correspondingly higher, indicating that changes in circulating and/or local activin A will affect resident macrophages through the expression of co-stimulatory molecules, thereby affecting the immune environment of the testes [118].

The characteristics of M2 cells in the testes are different from other tissues. In *rat* testes, after inflammatory stimulation or infection, in comparison with the liver, it will not cause classical polarization of M1 macrophages in the TMs. Instead, there is a weaker expression of pro-inflammatory factors IL-1b, TNF-a, and IL-6 but an increased expression of immuno-suppressive factors such as TGF-β1 and IL-10 than that of liver macrophages [119,120]. Bhushan et al. demonstrated that the M2 macrophages in *rat* testes have a limited response to inflammatory stimuli, since they can induce Treg differentiation in vitro and constitutively produce anti-inflammatory cytokines such as IL-10 and TGF-β1 in response to classical (IFN-γ + LPS) and alternative (IL-4) activation pathways. These macrophages can also inhibit Th1/ Th2 cell-mediated immune response [105,121,122]. Bergh et al. found that by ablating TMs, the inflammatory response induced by *human* chorionic gonadotropin was enhanced in the testes of *rats* with depleted TMs, speculating that this is one of the mechanisms by which TMs inhibit the local inflammatory response [123].

At the molecular level, the mechanism by which the M2 phenotype maintains immunosuppression may be as follows. Bhushan et al. have shown that macrophages in the *rat* testes can trigger the MAP kinase (p38 and ERK1/2) signaling pathway, which ultimately leads to the expression of AP-1 and CREB-dependent genes, and specific inhibitors on MAP kinase p38 and ERK1/2 can result in the inhibition of AP-1 and CREB signaling pathways, reducing the secretion of typical proinflammatory cytokine TNF-α [124]. At the same time, the low expression of TLR signaling pathway genes and the inhibition of the proinflammatory NF-κB pathway by attenuating the kinetics of IκBα degradation reduced the TMs response to inflammation or infectious stimuli [121,124]. In addition, M2 macrophages can inhibit immunity by activating the STAT3 signaling pathway [121]. AMPK activation also promotes the differentiation of macrophages to the anti-inflammatory M2 phenotype [125].

In summary, under normal physiological conditions, the immune privilege provided by M2 macrophages and other cells for the testes can protect against harmful autoimmune reactions caused by autoantibodies.

II. The M1 phenotype mainly plays a pro-inflammatory effect

In contrast, M1 macrophages have a pro-inflammatory effect. As mentioned above, the number of normal testicular M1 macrophages is very small (less than 20%), which may be related to the rarity of infectious inflammation of the testes. Once this occurs, the number of M1 macrophages significantly increases. Wendy et al. found that in the testes of *rats* that received LPS injection, the number of M1 macrophages reached three times the normal level, exceeded the M2 macrophages in the testes, and reached a peak 12 h after the injection [20]. Tung et al. showed that about 30% of F4/80-marked TMs in normal *mice* testes are MHC II positive, while in inflamed testes, almost all F4/80 cells are also MHC II positive, suggesting that macrophages are involved in the presentation of antigenic peptides [67]. In frozen sections of testes of *mice* infected with UPEC, only an increase in the number of interstitial F4/80-positive mononuclear phagocytes was observed, and no increase in the number of F4/80-positive cells in the intratubular was observed [126].

Experiments have shown that compared to peritonitis, the response of TMs is much weaker during testicular inflammation caused by pathogenic *E. coli* infection. Experiments by Bhushan et al. showed that compared to peritoneal macrophages (PMs) (5.3%), *rat* TMs only mobilize 1.2% of the total genome [127]. Genes related to both anti-inflammatory and pro-inflammatory response continue to be upregulated in testes, which cannot be seen in PMs [127]. This fact may reflect a delicate balance and fine-tuning of the testes between maintaining immune privilege and antibacterial response.

It is known that viruses such as HIV, HBV and the mumps virus can enter the testes and cause viral orchitis [128]. Recent studies have shown that coronavirus can also cause changes in immune cells in the testes [129]. A study conducted on testicular specimens of six patients who died of SARS-CoV infection showed that this virus can induce *human* orchitis [129]. However, the entry and diffusion of SARS-CoV-2 into host cells depend on the expression of angiotensin-converting enzyme 2 (AC2E) and transmembrane serine protease 2 (TMPRSS2), yet the co-expression of ACE2 and TMPRSS2 genes is only reported in spermatogonial stem cells and elongated sperm cells. Moreover, the existence of these receptors does not prove that the testes provide the site of viral infection; thus, TM is not a direct target of SARS-CoV [130,131,132]. Xu et al. found that compared with the control group, the number of CD3^+^ T lymphocytes and CD68^+^ macrophages (i.e., M1 macrophages) in the testicular interstitium increased. These cells may affect the function of Leydig cells to destroy the blood–testis barrier (BTB). They also secrete inflammatory cytokines and activate the autoimmune response in the *human* testes manifested by the deposition of large amounts of IgG in the germ cell [129]. In vitro experiments showed that the treatment of *human* peripheral blood mononuclear macrophages with SARS-CoV spike protein (S protein) could lead to the activation of the mononuclear macrophages and induce the secretion of cytokines and chemokines such as IL-6 and IL-8 [133]. This activation may be another important reason for severe orchitis caused by SARS-CoV, in addition to the decreased sperm motility caused by vasculitis and fever and the direct entry of SARS-CoV through ACE2 receptors expressed by Leydig cells [134,135].

Although the testes enjoy immune privileges, they cannot be completely isolated from the immune system. The EAO model induced by antigen immunization is used to clarify the pathological mechanism of autoimmune orchitis [136]. Once the sperm antigen destroys the immune privileges of the *rat* testes, the number of M1 macrophages in the interstitium mainly comes from the circulating CD68^+^ CD163^−^ monocytes of the testes, which gradually increases [137]. The TMs of EAO *rats* show higher levels of IFN-γ. Thus, they are involved in the complex production of inflammatory mediators such as TNF, IL-6, MCP-1 and NO at the same time (59). These inflammatory mediators can activate the autoimmune response and destroy the seminiferous epithelium, leading to *rat* autoimmune orchitis [137,138]. Therefore, TMs are involved in the occurrence and development of EAO, and M1 macrophages are the main pathogenic subgroup. M1 macrophages stimulate the immune response through the production of pro-inflammatory cytokines and antigen presentation, thereby activating T cells in target organs [137].

Studies have shown that TMs activate inflammatory signaling pathways through TLR ligands, inducing the expression of inflammation-related genes and producing inflammatory factors such as IL-1α and IL-1β, IL-6, IL-12, NOS2, activin A and decreasing production of TNF-α [99,139,140,141] Sarkar et al. found that IL-1 is an important inflammatory factor in *rat* testes, which can regulate the permeability of the blood–testis barrier (BTB) (134). IL-6 can promote B cells to differentiate and secrete antibodies and can directly or indirectly enhance the killing activity of natural killer cells (NK) and cytotoxic lymphocyte cells (CTL). In vitro experiments by Theas found that IL-6 can also induce spermatogenic cell apoptosis in *rats* [142].

In general, during acute or chronic testicular infection, M1 macrophages migrate and infiltrate into the testes, which changes the original M2/M1 macrophage balance and mediates the inflammatory response by releasing a large number of inflammatory factors. The local immune balance in the testes is broken, and the immune privilege is threatened. From this disruption, we conclude that two different types of macrophages in the testes respectively play a relevant role in anti-inflammatory and subdued proinflammatory responses (Figure 3).

#### 4.2.2. Macrophages in Epididymis

##### The Classification and Distribution of Macrophages in the Epididymis

In the epididymis, studies have shown that the location of *human* intra-epithelial macrophages is similar to that of *rodents* [9,139]. *Mouse* epididymal macrophages are located in all epididymal segments. According to the results of flow cytometry, Voisin et al. found that, similar to the testes, macrophages with the marker F4/80 or CD11b, CD64 ^+^ MHC II are the most abundant immune cells in the *mouse* epididymis, respectively accounting for 9.4% and 3.15% of all living cells in the caput and cauda [83]. In *human* and *mouse* epididymal MPS, the ‘macrophage’ phenotype is mainly defined by the expression of CD64, while dendritic cells are defined by the expression of CD11c and lack of CD64 [27,143,144,145]. *Mice* epididymal macrophages express CD64 most abundantly in the IS and then gradually decrease from the proximal end to the distal end of the epididymis [82,145]. Macrophages and DCs in the male reproductive system are often difficult to distinguish. A more accurate method to make this distinction is to use CD88 and CD89 for monocytes in *human* blood and tissues and to use HLA-DQ and FcεRIα for cDC2 for specific identification [146].

However, previous studies have shown that in *mouse* epididymis, basal cells (BCs) have macrophage-like characteristics, which is easy to confuse. Shum et al. used KRT5, F4/80, and CD11c to label *mouse* epididymal BC, macrophages, and DCs. It was found that peritubular DCs are the most abundant in IS and extend a large number of intraepithelial dendrites, mainly located on the epithelial side of the basement membrane, while in the caput and in all distal segments, peritubular DCs no longer project dendrites [84]. The difference between epididymal macrophages and BCs is that in IS, BCs are located at the base of the epithelium, and occasionally project a single, narrow extension toward the lumen, while each macrophage projects several slender intraepithelial dendrites. At the distal end of the epididymis (caput and cauda), however, BCs no longer emit dendrites, while macrophages emit short, intraepithelial dendrites at the proximal caput [84]. Additionally, in the caput, DCs are scattered at the base of the epithelium and do not seem to be interconnected, while BCs form a dense and continuous network of basal cellular bodies [84] (Figure 4). These results showed that BCs do not belong to the MP system despite their proximity and some morphological similarities with peritubular macrophages and dendritic cells [84].

##### The Function of Macrophages in the Epididymis

I. Epididymal macrophages maintain immune tolerance to sperm antigens

As mentioned earlier, macrophages and DCs in the male reproductive system are often difficult to distinguish. Battistone et al. observed that most macrophages in *mouse* epididymis with the ability to capture and process circulating antigens in all segments exhibited a “macrophage type” (CD64^+^), while a large proportion of macrophages with intrinsic and processed antigens are the “dendritic cell type” that express CD11c^+^ [82]. The exact role of macrophages in the epididymis remains unclear and may have similar functions to those in other mucosal systems, i.e., sampling the local environment, adjusting the immunogenicity and tolerogenicity of intraluminal antigens (including sperm antigens) and promoting the balance of epididymal homeostasis.

II. Epididymal macrophages play a role in monitoring and initiating inflammatory responses

The typical function of macrophages is the phagocytosis of phagocytose pathogens, dying cells, and cell debris. Therefore, the macrophage is often described as a “scavenger” of the immune system, mainly involved in maintaining innate immunity. The primary function of the epididymal macrophage is to actively maintain the integrity of the BEB [8]. *Mouse* macrophages located at the bottom of the ductus epididymis have the function of phagocytosis [147]. A study by Battistone et al. showed that in the *mouse* epididymis, RNA sequencing revealed that the “macrophage type” CX3CR1^+^ macrophage, which has the function of capturing and presenting antigens, is more abundant in the IS than in other parts of the epididymis [82]. Confocal microscopy showed that in the *mouse* IS, caput, and corpus, circulating antigens were internalized and processed by both the interstitial and intraepithelial MPS [82]. Following *E. coli* infection, *mouse* epididymal interstitial macrophages express less MHC II in the peritubular area and almost none in the epididymal epithelial macrophages [110]. This is consistent with the study of Battistone et al., wherein only interstitial MPS internalizes and processes antigens in *mouse* cauda, while intraepithelial MPS does not absorb antigens. These results indicate that all antigens have been captured before reaching the epithelial layer [82]. As such, there may exist a stronger effect of immune protection in the cauda.

After UPEC infection of *rodent* epididymis, epididymal epithelial cells recognize antigens through TLRs and produce cytokines, recruit and activate white blood cells, and they promote the production of defensins, thereby limiting infection [9]. Compared with the *rodent* testes, the expression of TLR 1–6 is higher in the testes, while the expression of TLR 7, 9 and 11 tends to be higher in the epididymis [9]. Cheng L et al. found that after infection of epididymis by *E. coli*, TLR4 and TLR5 on the epididymal head macrophages are activated. They then induce the production of pro-inflammatory cytokines through the classic inflammatory signal NF-κB pathway, whereas IFNA and IFNB expression is exclusively mediated by TLR4 [148,149]. UPEC can also activate *mouse* TLR11, which does not exist in *human* bodies [150]. Silva et al. reported different patterns of acute inflammation of the cauda induced by Gram-negative (LPS) and Gram-positive (LTA) bacterial products in *rat*, in which LPS can upregulate the mRNA levels of seven inflammatory mediators (Il1b, Tnf, Il6, Ifng, Il10, Nos2 and Nfkbia) and cause a strong inflammatory response. Conversely, LTA induced downregulation of one (Nfkbia) and upregulation of six (Il1b, Il6, Nos2, Il4 Il10 and Ptgs1) inflammatory gene transcripts, but increased the tissue concentration of three cytokines/chemokines (IL-10, CXCL2 and CCL2) [151].

The epididymis is also a target of viral infection. In the chronic simian immunodeficiency virus (SIV) infection model, SIV-infected macrophages and T cells were found at all levels of the reproductive tract, including in the epididymis. In the *mouse* model, however, macrophages that express CD45 ^+^ are the most likely target of HIV infection [152,153]. CD45 ^+^ macrophages are usually found in *human* epididymal epithelium and intestine, but rarely in the testes [154]. These cells may be the site where the early virus locates and stores HIV [155]. In addition to HIV, using a rhesus monkey model, Zeng et al. identified that CD68^+^ cells (macrophages/monocytes) are secret EBOV reservoir cells in the epididymal lumen but are usually absent during acute infections [156]. Based on this, a conclusion can be drawn that compared with other tissues, the epididymis may serve as a storage and replication site for certain viruses (such as HIV and EBOV), while in the testes or when confronting other viruses (such as SARS-CoV), the same function is not extant (Figure 5). We speculate that this is because of the difference of macrophage subtypes of the two organs and the target antigens between different viruses [66]. In *mice* with mild epididymitis, the infiltration is more common in the interstitial area, and a small amount of CD45^+^ cells aggregate in the lumen. However, in *mice* suffering from severe epididymitis, CD45^+^ macrophages mostly accumulate in the lumen rather than in the interstitium. Even in animals with the most severe epididymitis, no leukocyte infiltration was detected in the cauda [157].

## 5. Discussion

Compared with the testes, the epididymis is more likely to be attacked by external pathogens and lose its immune tolerance. This is because the immune environment of the epididymis and testes is different (Table 1). First, in terms of anatomy, the epididymis is more vulnerable to ascending infections of the reproductive tract, especially in the cauda. Moreover, the tight junction of the epididymis is much worse than that of the blood–testicular barrier (BTB). Second, the phenotype of macrophages is different in the epididymis and testes. For example, epididymal macrophages express infection targets of certain pathogens (HIV infection target CD45), while testicular macrophages do not. Therefore, the chance of pathogen invasion and infection of the epididymis is higher than that of the testes [158].

Moreover, there are differences in the level of immunity in different segments of the epididymis, and differences exist between *humans* and *rodents*. In the initial segment (IS) and the caput, macrophages are most abundant, and the dendric extension into the lumen is longer and more obvious. The epididymal body has the dual task of establishing immune tolerance to descending sperm and resisting retrograde pathogens. In the cauda, the phagocytosis, processing, and presentation of antigens are the most powerful. This change is related to the caput responsible for capturing sperm antigens, while the epididymal tail is responsible for preserving sperm and resisting ascending infections. Research into the immune status between MPS and different segments of the epididymis remains a future direction.

Due to the lack of effective markers to distinguish between macrophages and DCs in previous studies, these two types of cells have generally been referred to as MPS in some articles or are even considered to be one and the same. For this review, we conclude that testicular or eDCs can be distinguished by Ox62, CD11c, HLA-DQ and FcεRIα (DC) and CD64 (macrophages). This subdividing of the different subtypes of these two types of cells is essential for studying the immune mechanism of the testes and epididymis. The expression of co-stimulatory molecules is the biggest feature between different subtypes of MPS cells. For testicular macrophages, the maintenance of the anti-inflammatory phenotype is related to the level of hormones in the testes. The close relationship between DCs and Sertoli cells may be related to maintaining the DC phenotype. At the molecular level, the inhibition of the NF-ĸB signaling pathway and the activation of the STAT3 signaling pathway are related to the maintenance of immunosuppression. A “repolarization” process manifests when inflammation occurs. Moreover, the dualism that simply divides macrophages and DCs into two subtypes may not be rigorous or complete. Semi-mature phenotype DCs, double cDC1/2 marker DCs, and macrophages expressing the two cell markers of M1/2 suggest that the process of immune regulation in the testes is not simply an anti-inflammatory-pro-inflammatory opposition. Instead, there is a concurrent intermediate stage. Future research may focus on the finer immune regulation of the male reproductive system.

## Figures and Tables

**Figure 1 ijms-24-00053-f001:**
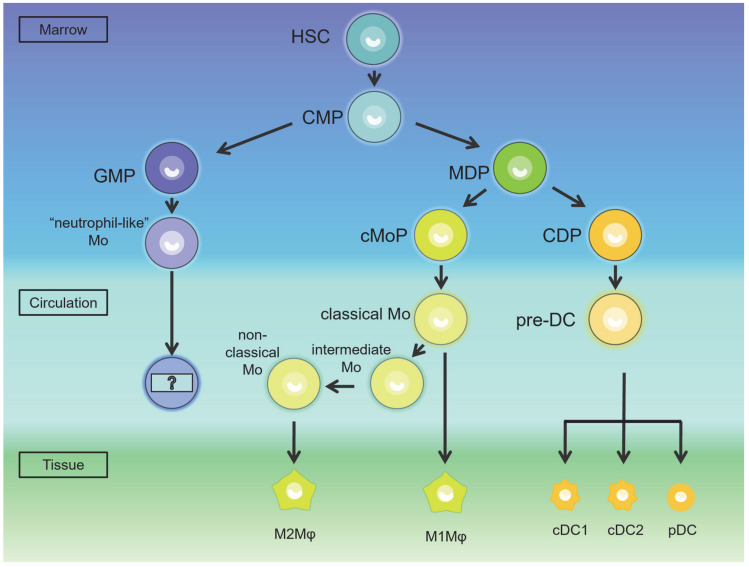
The generation of dendritic cells (DC) and macrophages (Mφ). Hematopoietic stem cell (HSC) first differentiates into myeloid progenitor (CMP). CMP differentiates into monocyte–dendritic cell progenitor (MDP) and granulocyte–monocyte progenitor cell (GMP). MDP produces two subsets, common monocyte progenitor (cMoP) and common DC progenitors (CDPs). cMoP produces classical monocytes. Classical monocytes can then generate into intermediate and non-classical monocytes. In tissues (epididymis and testes), M1 (classical) macrophages are differentiated from classical monocytes, while M2 (non-classical) macrophages are differentiated from non-classical monocytes. CDP can differentiate into conventional DC1 (cDC1), conventional DC2 (cDC2) and plasmacytoid DC (pDC) through pre-DCs. GMP produces a subset of “neutrophil-like” monocytes. (HSC: hematopoietic stem cell; CMP: myeloid progenitor; MDP: monocyte–dendritic cell progenitor; GMP: granulocyte–monocyte progenitor cell; cMoP: common monocyte progenitor; CDP: common DC progenitor; cDC1: conventional DC1; cDC2: conventional DC2; pDC: plasmacytoid DC; DC: dendritic cell; Mφ: macrophage; Mo: monocyte).

**Figure 2 ijms-24-00053-f002:**
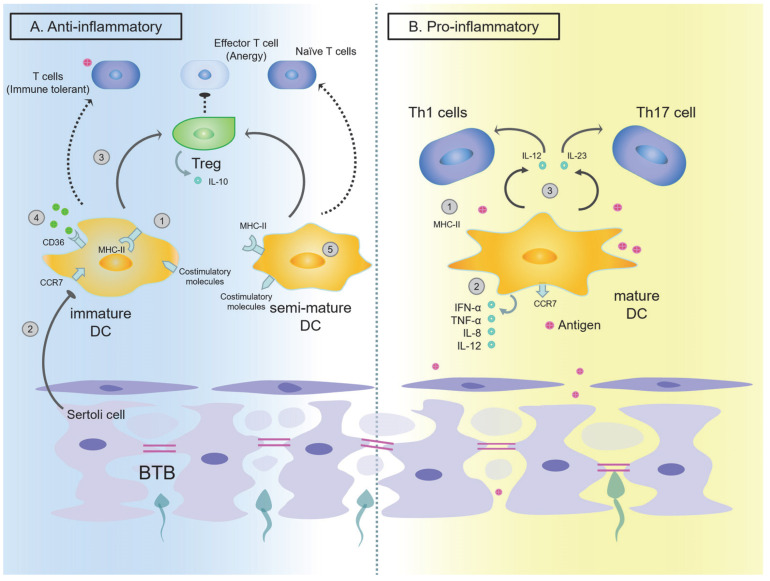
Regulation of testicular DC in physiological conditions (mainly anti-inflammatory) and pathological conditions (mainly pro-inflammatory). (**A**) Under physiological conditions, the DCs in the testes are mainly immature DCs and semi-mature DCs, maintaining an anti-inflammatory state. ① DCs in normal testes do not lack MHC II or co-stimulatory molecules. Inducing T cell anergy and producing Treg absolutely require the presence of co-stimulatory molecules in DCs ② Sertoli cells lead to downregulation of co-stimulatory molecules on the surface of DCs. ③ DCs in normal testes can induce the production of regulatory T cells (Treg) and together with Treg produce the anti-inflammatory cytokine IL-10. In addition, they secrete T cell development regulator IDO. ④ Immature DCs have strong internalization ability and can take up tissue fragments or apoptotic cells through CD36 or integrin. ⑤ Under physiological conditions, semi-mature phenotype of DCs (which may be the majority in normal testes) expresses MHC II and co-stimulatory molecules, but cannot stimulate the proliferation of naive T cells, but can induce the production of Treg. (**B**) Under pathological conditions, testicular DCs transform from immature DCs to mature DCs, mainly for pro-inflammatory regulation. ① When DCs are attacked by antigens, the expression of chemokine receptors CCR7 is upregulated and CCR2 is downregulated, which marks DCs transformation into a mature phenotype. ② Mature DCs produce high levels of pro-inflammatory cytokines, such as IFN-α, TNF-α, IL-8 and IL-12. ③ Mature DCs produce IL-12, IL-23. IL-12 is able to predispose activated T cells to differentiate into an inflammatory Th1 phenotype, whereas IL-23 is crucial for terminal differentiation and survival of Th17 cells. (DC: dendritic cell; BTB: blood–testicular barrier; Treg: regulatory T cell).

**Figure 3 ijms-24-00053-f003:**
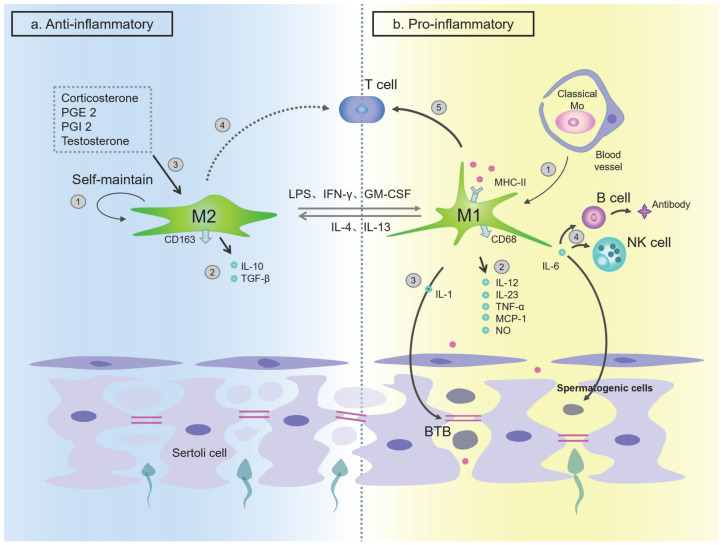
Regulation of testicular macrophages in physiological conditions (mainly anti-inflammatory) and pathological conditions (mainly pro-inflammatory). (**a**) Under physiological conditions, the testes are dominated by M2 macrophages, which exert anti-inflammatory and immune tolerance functions. ① Under normal conditions, M2 can independently maintain itself. ② M2 macrophages mainly secrete anti-inflammatory cytokines such as IL-10 and TGF-β. ③ The M2 anti-inflammatory phenotype is maintained by corticosterone, PGE 2, PGI 2, and testosterone in the surrounding interstitial fluid (IF). ④ Due to the lack of co-stimulatory molecules, the antigen presentation process of M2 macrophages to T cells is blocked. (**b**) Under pathological conditions, M1 macrophages become the quantitatively dominant group and mainly play a pro-inflammatory function. ① Classical monocytes are recruited to the testes and differentiate into M1 macrophages. ② M1 macrophages mainly secrete pro-inflammatory cytokines such as IL-1, IL-6, IL-12, IL-23, TNF-α, MCP-1, and NO. ③ Among them, IL-1 is an important inflammatory factor, which can regulate the permeability of the blood–testicular barrier (BTB). ④ IL-6 can promote B cell differentiation, T cell antibody secretion, and directly or indirectly enhance the killing activity of natural killer cells (NK cells); in vitro experiments have found that IL-6 can also induce spermatogenic cell apoptosis. ⑤ M1 macrophages are MHC II positive, thereby participating in the presentation of antigens to activate T cells. In the testes, lipopolysaccharide (LPS) or IFN-γ, GM-CSF can promote the polarization of M2 macrophages to M1 macrophages, and IL-4 and IL-13 can induce M1 macrophages to polarize to M2 macrophages. (Mo: monocyte; BTB: blood–testicular barrier; NK cell: natural killer cell; LPS: lipopolysaccharide).

**Figure 4 ijms-24-00053-f004:**
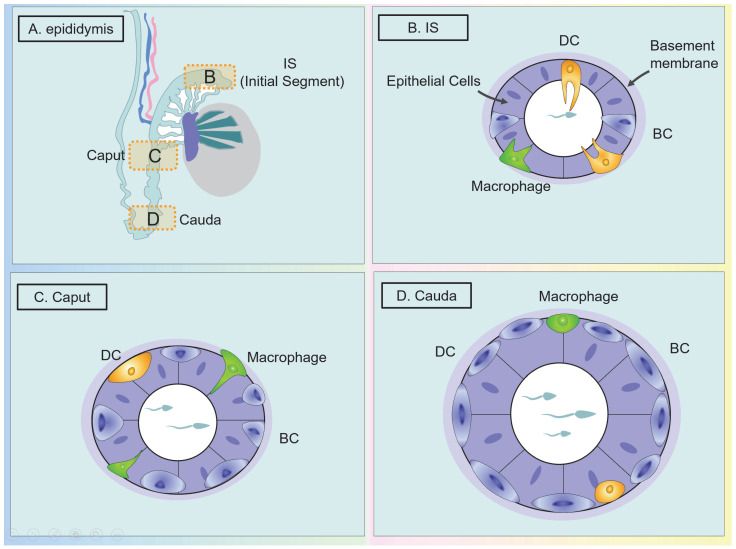
Morphological changes of immune cells in different segments of the epididymis. Marking mouse epididymal BCs, macrophages, DCs with KRT, F4/80 and CD11c. (**A**,**B**) In the IS, the BCs are located at the base of the epithelium and occasionally send out single, narrow dendrites into the lumen. However, each macrophage and peritubular DC extended a large number of intraepithelial dendrites, mainly located on the epithelial side of the basement membrane. BCs show a characteristic pyramid shape, while the dendrites of DC are slender. The location and morphology of macrophages in the epididymis are similar to DCs. In *human* epididymis, unlike in mouse epididymis where a dense network of DCs is localized in the epithelium, DCs are identified in the interstitial compartment of the epididymis (not shown). (**C**) In the caput and beyond, BCs and DCs no longer extend out of dendrites. Only macrophages emit short dendrites at the proximal end of the caput. (**D**) In the cauda, DCs are scattered at the bottom of the epithelium, while BCs form a continuous network. (BC: basal cell; DC: dendritic cell; IS: initial segment).

**Figure 5 ijms-24-00053-f005:**
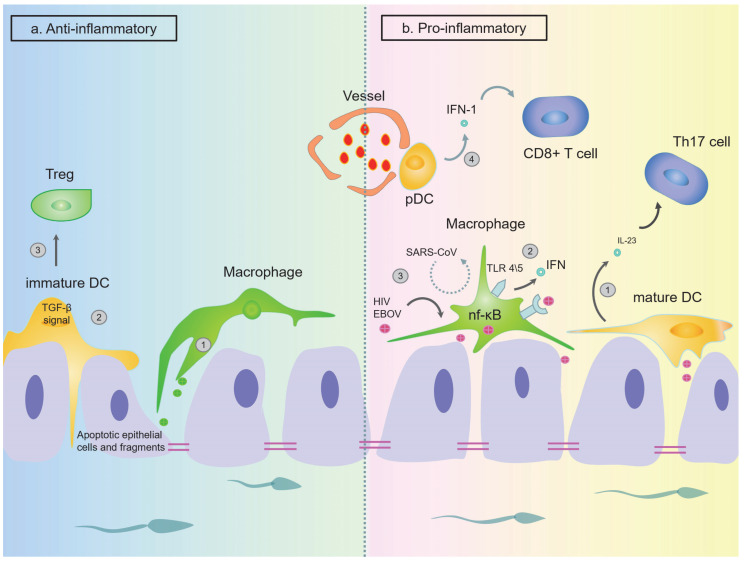
Physiological conditions (mainly anti-inflammatory) and pathological conditions (mainly pro-inflammatory) regulation of epididymal immunity. (**a**) Under physiological conditions, DCs and macrophages mainly exert immune tolerance effects. ① DCs and macrophages can phagocytose the apoptotic epithelial cells and their fragments. Together, they maintain the integrity of the blood–epididymis barrier (BEB). ② DCs can maintain the immature state through TGF-β signal transduction to maintain immune tolerance. ③ DCs can inhibit the inflammatory response by promoting the differentiation of Treg cells. (**b**) In pathological conditions (such as inflammation), DCs and macrophages begin to exert pro-inflammatory functions. ① When chronic epididymitis occurs in *humans*, DCs produce the majority of IL-23, which is believed to play a key role in the terminal differentiation and survival of Th17 cells. ② After the pathogen (such as *E. coli*) infects the epididymis, the level of MHC II expression increases, and TLR4 and TLR5 on the macrophages are activated, which induce the production of pro-inflammatory cytokines such as IFN through the classical inflammatory signaling NF-κB pathway. ③ The epididymis may serve as a site for the storage and replication of certain viruses (such as HIV and EBOV) when they are infected. However, this does not occur during the invasion of other viruses (such as SARS-CoV). This is related to antigen specificity. ④ pDCs, which are found only in chronic epididymitis, are arranged as clusters around the small vessels or as single cells. It is a special antiviral cell that can rapidly secrete high levels of IFN-I and facilitate and amplify the cytotoxic T cell functions. (DC: dendritic cell; BEB: blood–epididymis barrier; Treg: regulatory T cell).

**Table 1 ijms-24-00053-t001:** T Cell-surface markers used to identify MPS in testes and epididymis.

Location	Cell Type	Classification	Markers		References
			Human	Mice	
Blood	Monocytes	Classical Monocytes	CD14^++^CD16^−^	Ly6Chi CX3CR1int CCR2^+^ CD62L^+^ CD43low	[15,17]
		Nonclassical Monocytes	CD14^+^CD16^++^	Ly6Clow CX3CR1hi CCR2low CD62L^−^	
		Intermediate monocytes	CD14^++^CD16^+^	Ly6C^++^ CD43^++^	[15]
Testes	DC	cDC	CD1C^+^	CD11b^+^ CD172a^+^ (cDC2)	[16,30]
			CD141^+^	CD8α^+^ CD103^+^ (cDC1)	
	Macrophage	Peritubular macrophages	CD68^+^ CD163^+^	M-CSFRlo CD64lo MHCII^+^	[20,24,26,74,112]
		Interstitial macrophages		M-CSFR^+^ CD64hi MHCII^−^	
		M1 phenotype (infiltrating TMs)	MHC II^+^ CD68^+^ CD80^+^ CD86^+^	CD68^+^ CD163^−^ (rat)	[18,20]
		M2 phenotype (resident TMs)		CD68^−^ CD163^+^ (rat)	
		Intermediate TMs		CD68^+^ CD163^+^ (rat)	
Epididymis	DC	cDC1	CD11c^+^ IL-23p19^+^	CD64^−^ CD24^+^ CD11c^+^ MHC II^+^ CD11b− CD103^+^	[53,80]
		cDC2		CD64^−^ CD24^+^ CD11c^+^ MHC II^+^ CD11b^+^ CD103^−^	
		pDC	Lin-MHC II^+^ CD303(BDCA-2)^+^ CD304 (BDCA-4)^+^		[15,53,101]
	Macrophage		F4/80 or CD11b^+^ CD64 ^+^ MHC II		[83]

## Data Availability

Not applicable.

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
