# Peer review of "The Role of Mononuclear Phagocytes in the Testes and Epididymis"

_ijms, 2022, doi:10.3390/ijms24010053_

Round 1
Reviewer 1 Report
It is an interesting review discussing the role of the mononuclear phagocytic system (MPS) in the male reproductive tissues, in particular testis and epididymis, that maintains the balance of pro-inflammatory and immune tolerance. The authors summarized and discussed very well the bibliography available in both organs. However, there are parts of the review that are confusing, for example: when the authors are describing what is known in the testis/epididymis or if they are writing about a general concept that is known in other organs/systems but is unknown in the male reproductive tract.
- The following sentence in the abstract is not correct: “The mononuclear phagocytic system (MPS) is the primary innate immune cell group in male reproductive tissues, maintaining the balance of anti-inflammatory and immune tolerance”. The balance should be between pro-inflammation and tolerance.
- Why did the authors repeat the keyword “MPS” in the PubMed search?
- The authors wrote in the abstract line 25 “our study showed …” but this is a review, and no new results are discussed in it.
- For this reviewer, there is not enough evidence in the epididymis that indicates the following statement: “In the testis and epididymis, MPS cells are mainly suppressed subtypes (M2 and cDC2) under physiological conditions, which maintains the local immune tolerance”
- In Line 45, it should be written pro-inflammatory instead of anti-inflammatory.
- Monocytes can differentiate into macrophages or dendritic cells in different organs, but that concept is not discussed in the review.
- In paragraph 4.1.1.2, lines 221-225, the authors are describing a general concept that is unclear in the epididymis/testis. They should clarify that important point.
- The sentence in lines 356 -358 should be redone.
- In paragraph 4.1.2.2, lines 386-388, are the authors describing a general concept? They should clarify that. In the epididymis, it is known that CD103- cells express low levels of co-stimulatory molecules.
- The title of 4.2.1. should be macrophages instead of DCs.
- In line 680, the authors should include some references about epididymis together with references 143 and 144.
- Line 727 - 730 needs to be re-written, it is not clear what the authors wanted to highlight.
- Line 737 needs to be rewritten.
Figures
- legend Figure 2: for the authors, are peritubular macrophages the interstitial ones? Those macrophages do not extend dendrites into the lumen.
- legend Figure 2: “Due to the particularity of the epididymal location, the DCs observed in the cauda have a strong ability to internalize and process antigens, while macrophages possess a strong ability to capture and process antigens”. It is missing the region that the authors are describing.
- Figure 3 (related to the epididymis) should be after Figure 4 (related to the testis).
Reviewer 2 Report
This article in an interesting review concerning the role of mononuclear phagocytes in the testis and epididymis.
However, despite a global overview of these cells, the authors distinguish two cell types (dendritic cells and macrophages) in two organs (testis and epididymis). If they cite the halo cells, nothing about lymphocytes present in the epididymal epithelium.
Moreover the text mixes data obtained in animal models and in humans. It would be necessary to distinguish between what is detected in mice and what is actually identified in humans. In this sense, Table 1 is very informative. It should be the same in the text. The organization of the plan is too burdensome and should be reviewed to be more clear for the readers.
Round 2
Reviewer 2 Report
The manuscript has been improved